# Optimization of Additively Manufactured and Lattice-Structured Hip Implants Using the Linear Regression Algorithm from the Scikit-Learn Library

Rashwan Alkentar [1],* and Tamás Mankovits [2]

1   Department of Mechanical Engineering, Faculty of Engineering, University of Debrecen, Ótemető u. 2-4.,
    H-4028 Debrecen, Hungary
2   Doctoral School of Informatics, Faculty of Informatics, University of Debrecen, Kassai u. 26.,
    H-4028 Debrecen, Hungary
*   Correspondence: rashwan.alkentar@eng.unideb.hu

**Abstract:** As the name implies, patient-specific latticed hip implants vary in design depending on the properties required by the patient to serve as a valid suitable organ. Unit cells are typically built based on a 3D design of beams, and the properties of unit cells change depending on their geometries, which, in turn, are defined by two main parameters: beam length and beam thickness. Due to the continuous increase in the complexity of the unit cells' designs and their reactions against different loads, the call for machine learning techniques is inevitable to help explore the parameters of the unit cells that can build lattice structures with specific desirable properties. In this study, a machine learning technique is used to predict the best defining parameters (length and thickness) to create a latticed design with a set of required properties (mainly porosity). The data (porosity, mass, and latticed area) from the properties of three unit-cell types, applied to the latticed part of a hip implant design, were collected based on the random length and thickness for three unit-cell types. Using the linear regression algorithm (a supervised machine learning method) from the scikit-learn library, a machine learning model was developed to predict the value of the porosity for the lattice structures based on the length and thickness as input data. The number of samples needed to generate an accurate result for each type of unit cell is also discussed.

**Keywords:** machine learning; unit cells; finite element analysis; optimization; hip implant





## 1. Introduction

Lattice structures are mainly layers of beam elements connected to each other to form a latticed body, hence allowing the designers to distribute the flow of material in a structurally effective way based on the required design [1]. Due to their high strength-to-weight ratio and many other interesting properties, lattice structures have attracted considerable interest in the field of structural design. The idea of a lattice structure's ease of geometrical control and adjusting properties has pushed its importance to a high level [2–4]. Several studies have tried to use this feature to change the structure's properties into field-suitable structures. Abdulhadi and Mian attempted to adjust the properties of the lattice structures to suit a biomedical application by manipulating the lattice's defining parameters [5]. Other studies tried to adjust the parameters to find the most robust structure [6] to improve bone ingrowth when lattices were applied to medical implants [7] or to reduce the stiffness of the structure [8].

However, adjusting the unit cells' parameters increases the complexity of the design and, consequently, the difficulty in manufacturability. Additive manufacturing (AM) emerged to solve this problem by reducing the constraints from the manufacturing point of view [9–12]. To date, AM techniques can manufacture lattice structures with the little specific details and small features they might contain [13–16]. Gabriele Cortis [17] performed

an accurate study of a parametric finite-element model of a Ti6Al4V alloy implant and demonstrated how the conventional implant could be redesigned with lattice structures applied to the trabecular structure with the purpose of reducing the stress shielding effect and then manufactured using AM technology.

Recent studies have tried to produce perfectly suitable unit-cell structures by adjusting their parameters. Tailored designs of lattices can be produced to match bone performance, and it can be proven that stiffness and collapse load on the latticed design depend on its density and geometry [18,19]. Meiling Fan created a novel lattice structure with a graded lattice core to achieve a gradient design through a tailored porosity. The study was coupled with an analytical model to determine the compression response of graded lattice cores through additive manufacturing and compression tests [20]. Researchers believe there is still a wide range to be explored in the latticed designs that might provide a better performance in comparison to the already-built structures. This fact has called for machine learning (ML) to step in since it can process topological optimizations on the parameters of the unit cells, handle complex numerical analyses, and produce in a wide range of options. Topology optimization has been applied in the field of unit cells for a long time; yet, it is time-consuming due to the complex coding needed [21].

The inverse design or "inverse problem", a collection of new research methods used in natural sciences and engineering where the particular configuration of a geometry material or process is determined starting from the targeted search activity [22], is used as a method to achieve the required properties within the unit cells. However, it is still challenging and difficult to accomplish due to the complex nonlinearity between the lattice parameters and their resulting mechanical properties as Guoji Yu et al. explained when proposing a deep learning-based strategy to design lattice structures with customized mechanical behavior. The strategy involved the design and manufacturing of heterogeneous lattice structures using octet-truss and rhombic dodecahedron cells, and then training an artificial neural network to predict the mechanical properties based on the finite-element analysis data [23]. Jier Wang and Ajit Panesar proposed a novel lattice generation approach to design graded lattice structures with the help of machine learning. The study constructed a neural network (NN)-based inverse lattice generator developed to provided unit cells as an output from the input being the target mechanical properties. They concluded that ML could enhance the design optimality and speed for the lattice structure's optimization [24]. Shuai et al. collected 57 samples from previous studies and used the support vector regression (SVR) method to predict the mechanical properties of lattice structures. The input data were porosity, elastic modulus, material density, and unit length of the lattice unit. The research provided a method able to promote the design process of the unit cells by predicting the mechanical properties effectively [25]. Sangryun Lee et al. used a deep learning method to study the optimized shape of the beam elements. Then, combined with a hybrid neural network and genetic optimization adaptive method, the study was able to generate superior lattice structures. The resulting optimized designs were manufactured via AM and evaluated through compression testing. The validation results were better regarding the modulus and strength [1].

Adithya et al. proposed a development by introducing an innovative inverse machine learning framework that harnessed the power of generative adversarial networks (GANs). This framework was specifically designed to streamline the optimization of lattice unit cells, aiming to produce specific mechanical properties. Their approach represents a significant leap forward in material engineering and design as it allows for the precise adjustment of lattice structures to meet the desired performance criteria [26]. Guoji Yu and Weidong Song generated the stress–strain curves of lattice structures via the finite-element method (FEM) and then used the data to train an artificial neural network that was able to reproduce the prediction of the FEM. The design can broaden the design space of the generated lattice structures with the desired properties [23]. The approach of a controllable inverse design was also used to create designs for auxetic metamaterials. These are specially engineered

periodic composites characterized by a negative Poisson's ratio, a property not commonly observed in natural materials.

Additionally, these metamaterials exhibit electromagnetic properties that diverge from what is naturally occurring. This technique opens new avenues for adjusting materials with unprecedented and exotic mechanical and electromagnetic behaviors, resulting in new possibilities for applications in various fields of engineering [27,28], with the required Poisson's ratio and Young's modulus. The design can also be extended to many types of architected materials [29]. In [30], a deep learning approach was proposed for the reverse design of gradient mechanical metamaterials, attaining a multi-scale design based on the neural networks and topology optimization together. The design accelerated the emergence of a high-performance structure with a rapid design flow that took up to 2 seconds only, and could provide a reference for the topology optimization design of mechanical metamaterials. Chonghui Zhang et al. developed a hybrid design framework combining deep learning forward and inverse designs based on the mixture density network (MDN) with the purpose of designing low-porosity auxetic materials [31]. Zhongyuan Liao et al. used a deep neural network (DNN) to create a surrogate model that could represent the desired mechanical properties as a function of the geometric parameters after obtaining the data of already-calculated mechanical properties and geometric parameters [32].

Hany et al. designed diamond-shaped lattice structures with various strut lengths, diameters, and orientation angles. The manufacturing was performed via laser-power bed fusion (LPBP) with a Ti-64 alloy. The specimens were tested on compression to obtain the values of elastic modulus, specific strength, and ultimate strength. The authors also checked the optimized materials' properties based on a trained model of a finite-element analysis database generated with hundreds of thousands of geometries. The study used a shallow neural network (SNN), DNN, and deep learning neural network (DLNN). The model was designed to predict the mechanical properties using ML techniques, and the results showed that the DLNN performed the best with mean errors of 5.26%, 14.6%, and 9.39% for the ultimate strength, elastic modulus, and the specific strength, respectively [33].

Machine learning techniques were also applied to help predict the structural properties with better buckling resistance and optimal lattice unit cells [34]. Myriam et al. [35] optimized the stem of a hip implant with the help of machine learning techniques combined with a finite-element analysis. The research was performed with the purpose of reducing the stress shielding of the implant. An artificial neural network was used in combination with the finite-element analysis to optimize the hip stem, and it resulted in reducing the length of the stem neck, which helped reduce the stress shielding effect.

The final element was a supporting validation and data generation method for the ML techniques. In a study, Jan Hendrik et al. aimed at inverting the structure–property map of truss metamaterials using deep learning; the FEM was used to generate a dataset with millions of randomly generated lattices. This dataset greatly helped the deep learning method by providing good training, validating, and testing sets [36]. Qi Zhenchao et al. used the FEM to generate data that helped with the predictions of the elastic modulus and the shear modulus of carbon fibers [37]. The results from the FEM were compared with the results of the machine learning techniques to validate how well the model worked [26].

The purpose of this study is to use a supervised machine learning approach known as the linear regression method. The linear regression method was chosen based on the fact that the purpose of the model was to predict quantities, and the number of samples was over than 50 as stated by the map for choosing the estimator by the scikit-learn community [38]. A machine learning model was developed to predict the porosity of the lattice model of a hip implant as a dependent variable based on the two main independent variables, which were the unit cells' beam length and thickness. The study also investigates the number of learning points for each of the three unit-cell types needed to generate accurate results.

## 2. Materials and Methods

Using the design tool SpaceClaim, integrated within the ANSYS software 2021 R2 (Canonsburg, PA, USA) a hip implant was designed. Specific attention was dedicated to the stem component, where a lattice structure was ingeniously applied. The design was engineered with the purpose of enhancing the overall strength of the implant, the durability, and, most of all, the osseointegration capabilities. As shown in Figure 1: (a), a 2 mm-thick shell all over the transparent part of the implant body is to be latticed with three unit-cell types: (b) 3D-lattice infill, (c) double-pyramid lattice and face diagonals, and (d) octahedral lattice 2 (names of the unit-cell types are based on the Ansys SpaceClaim 2021 R2 Canonsburg, Pa, USA software classifications). From a medical point of view, the outer thickness of 2 mm was chosen to be latticed in order to ease the interaction between the bone and implant since the value of Young's modulus for the implant material decreased, as proven in a previous study [39], into a value close to the bone's, which made the integration smoother [40].

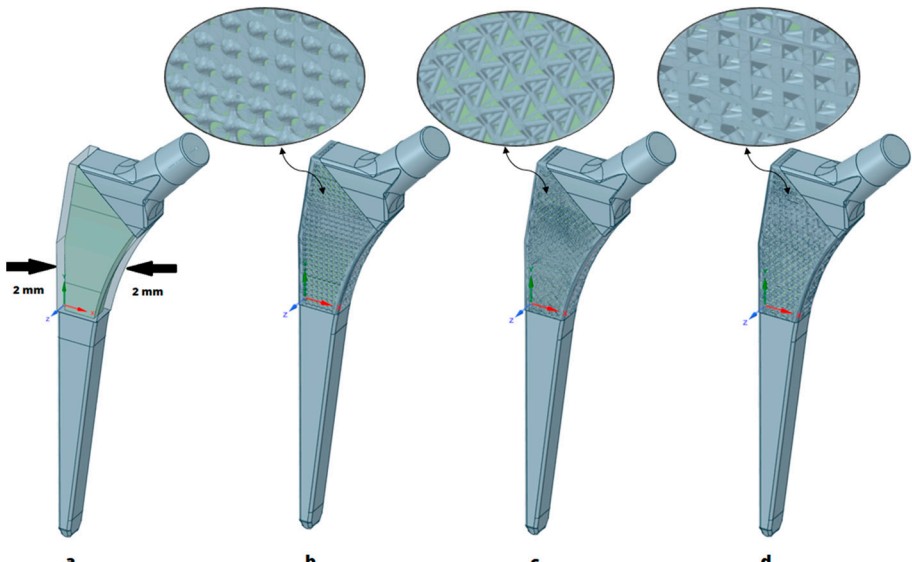

**Figure 1.** (**a**) Hip implant design, (**b**) 3D-lattice infill, (**c**) double-pyramid lattice and face diagonals, (**d**) octahedral lattice 2.

The porosity of lattice structures within the latticed part can be determined by employing a calculation that relies on the volume measurements, as shown in Equation (1). This equation serves as a tool for quantifying the degree of porosity within the lattice structures, providing valuable insights into their composition and potential mechanical properties. By using volume-based metrics, this approach offers a quantitative means to assess the material voids and density distribution within the lattice, which, in turn, can inform critical decisions in the design and optimization process.

$$\phi(\%) = \frac{V_{bulk} - V_{structure}}{V_{bulk}} \cdot 100 \tag{1}$$

where $V_{bulk}$ is the volume of the part before applying the lattice, $V_{structure}$ is the volume of the part after applying the lattice. The porosity was calculated based on a set of many values of the length and thickness of the lattice structure's strut for each type of unit cell. The samples were designed with a consideration of the manufacturing limitations; the thickness was at least 0.2 mm and the acceptable range of porosity was 50–90%, as clarified in [41]. The following Table 1 shows the range of values for each type. The selection of the beam's length and thickness was a pivotal decision and needed careful consideration, and their values were meticulously selected, taking into account a spectrum of values that ultimately resulted in a range of porosity values.

**Table 1.** Range of values for each type of unit cell.

| Unit-Cell Type | Length (mm) | Thickness (mm) | Porosity (%) |
|---|---|---|---|
| Three-dimensional-lattice infill | 1–1.6 | 0.2–1.5 | 50–83 |
| Double-pyramid lattice and face diagonals | 1.8–2.5 | 0.2–0.6 | 51–86 |
| Octahedral lattice 2 | 2–2.7 | 0.2–0.65 | 51–87 |

The input data can be visualized as shown in Figure 2.

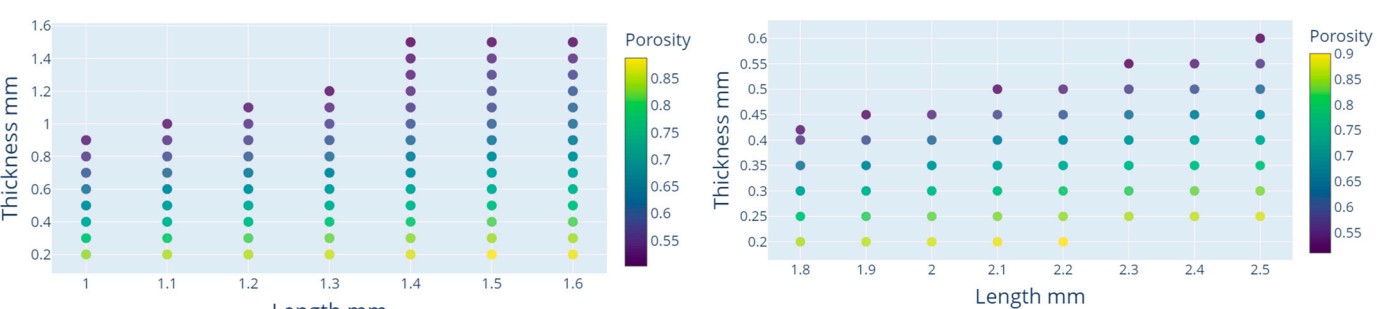

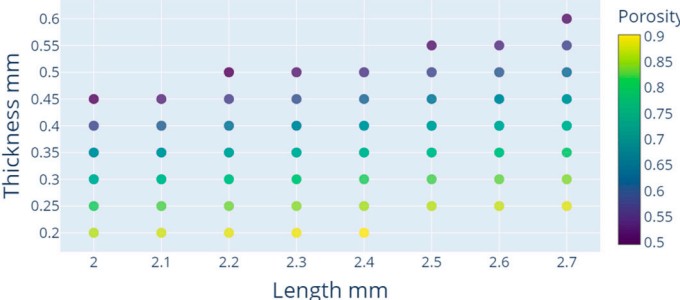

**Figure 2.** Visualized input data for all three types of lattice structures.

For a clearer visualization of the data chosen for the input of the model, histogram plots are shown in Figure 3 for all three types of chosen unit cells.

Linear regression is a supervised machine learning algorithm that helps predict continuous values based on the input variables. All methodologies for this linear regression algorithm were tried by Irwanda Laory [42] and were proved to be capable of predicting accurate results. The algorithm forms a linear relationship between the independent variables (beam thickness and length) and the dependent variable (porosity). Then, the algorithm investigates the best-fit line that minimizes the sum of the squared errors between the predicted and actual values, hence the name regression line. The equation of the regression takes the following form:

$$y = \beta 0 + \beta 1 * x1 + \beta 2 * x2 + \ldots + \beta n * xn$$

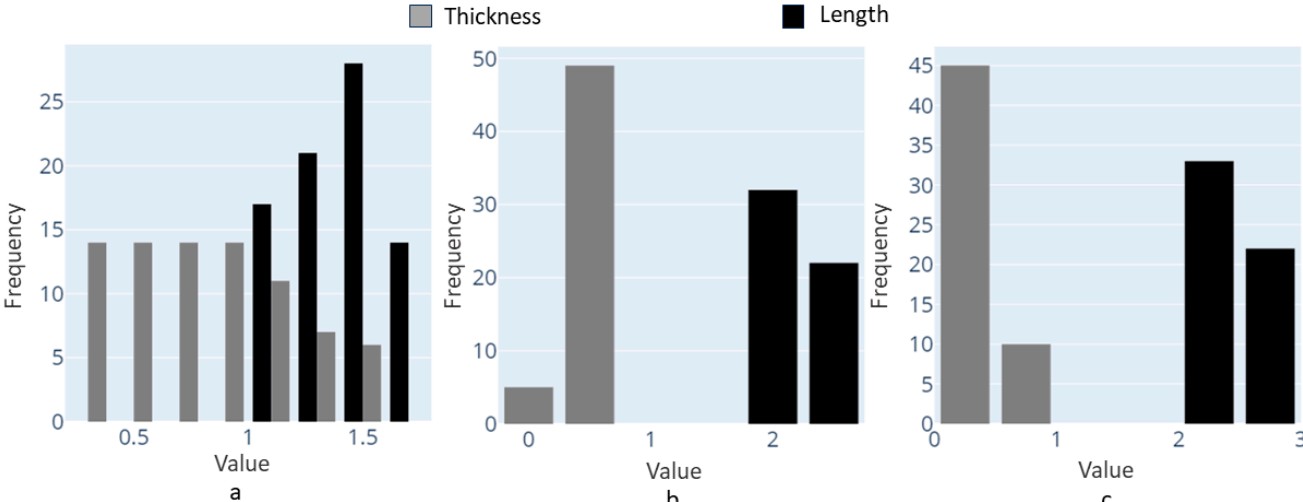

**Figure 3.** Histogram plots for data selection: (**a**) 3D-lattice infill, (**b**) double-pyramid lattice and face diagonals, (**c**) octahedral lattice 2.

Figure 4 shows the basic principle of the linear regression method and how the variables are calculated:

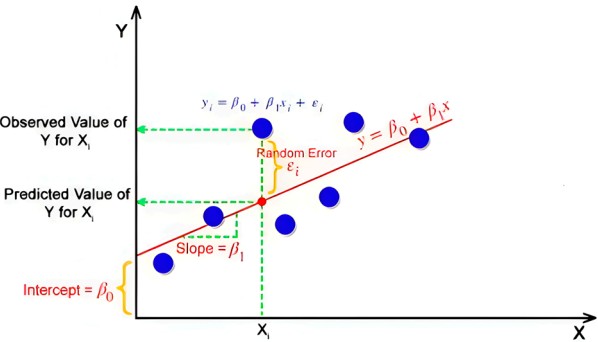

**Figure 4.** Basics of the linear regression method. Reprinted/adapted with permission from Ref. [43].

where y is the dependent variable, x1, x2, . . ., xn are the independent variables, β0 is the intercept, and β1, β2, . . ., βn are the coefficients. The linear regression algorithm aims at estimating the values of the coefficients (β0, β1, β2, . . ., βn) in a way that minimizes the sum of the squared errors. This procedure, followed by the scikit-learn library, is called the ordinary least squares (OLS) method [44].

The data were collected for the three types. Each piece of data contained the configurations of the length and thickness and the corresponding porosity. The necessary libraries were imported to the model: Pandas for data manipulation, NumPy for numerical operations, LinearRegression from scikit-learn [38] for the linear regression model, r2_score from scikit-learn for calculating the R-squared score, and plotly.graph-objs for creating the 3D plot. The dataset was loaded from a CSV file using Pandas in Jupyter Notebook 6.4.5. The features (length and thickness) and target variable (porosity) were extracted from the dataset and stored in arrays, 'X' and 'y', respectively. The linear regression model was created and trained using the feature array 'X' and the target variable 'y'.

The model was used to predict the values of porosity for the feature space. The feature space was created by generating a range of values for length and thickness using the NumPy's linspace function. A mesh grid was created using these ranges, resulting in 'length-mesh' and 'thickness-mesh', which represented all possible combinations of length and thickness values. The feature space was built by combining these two mesh grids using

np.column-stack. Finally, the model predicted the porosity values for the feature space, which were reshaped to match the shape of the mesh grid in 'porosity-pred'.

To assess the accuracy of the model, the R-squared function score was used. This metric assesses the predictive performance by comparing the estimated porosity values (X) to the actual porosity values (y). The R-squared score quantified the degree of correlation between the predicted and actual porosity values. In other words, the R-squared score is a statistical measure that represents the proportion of variance in the dependent variable (y) that is being predicted from the independent variable (X), where it ranges from 0 to 1, with 1 indicating that the model perfectly predicts the target variable based on the features.

In order to find out the optimal number of learning points required to construct a precise machine learning model, a thorough investigation was conducted across three distinct cases for each type of lattice structure. For each case, a machine learning model was developed using progressively increasing portions of the available data, beginning with one third, followed by two thirds, and ultimately encompassing the entire dataset as shown in Table 2.

**Table 2.** Number of learning points for each type of unit cell.

| Unit-Cell Type | Case 1 | Case 2 | Case 3 |
|---|---|---|---|
| Three-dimensional-lattice infill | 27 | 54 | 81 |
| Double-pyramid lattice and face diagonals | 18 | 35 | 54 |
| Octahedral lattice 2 | 18 | 35 | 54 |

## 3. Results

The machine learning models demonstrated a good performance across all three types of lattice structures, achieving acceptable levels of accuracy. In order to assess the accuracy, a thorough evaluation was conducted by comparing the predicted values with the actual data in the dataset. The detailed breakdown of these accuracy scores for each case is provided in Table 3.

Through the careful examination of the accuracy levels, it is noticeable that precision can be attained with a relatively modest number of learning points. Specifically, 81 learning points were sufficient for the 2D lattice infill, while 54 learning points were adequate for the double-pyramid lattice and Face Diagonals. Octahedral lattice 2 demonstrated a similar efficiency, also requiring 54 learning points to yield accurate results.

**Table 3.** Model's accuracy for each case of lattice types.

| Unit Cell Type | Case 1 Accuracy (%) | Case 2 Accuracy (%) | Case 3 Accuracy (%) |
|---|---|---|---|
| Three-dimensional-lattice infill | 98 | 96 | 96 |
| Double-pyramid lattice and face diagonals | 98 | 98 | 99 |
| Octahedral lattice 2 | 98 | 99 | 99 |

In Figure 5, a visual representation of the model's outcomes is presented. The actual porosity values are depicted as individual blue data points scattered across the plot. In contrast, the predicted porosity values are illustrated through a color-graded surface plot, providing a comprehensive visualization of the model's estimations across the entire feature space. This graphical representation exhibits a clear and intuitive understanding of how well the model aligns with the data.

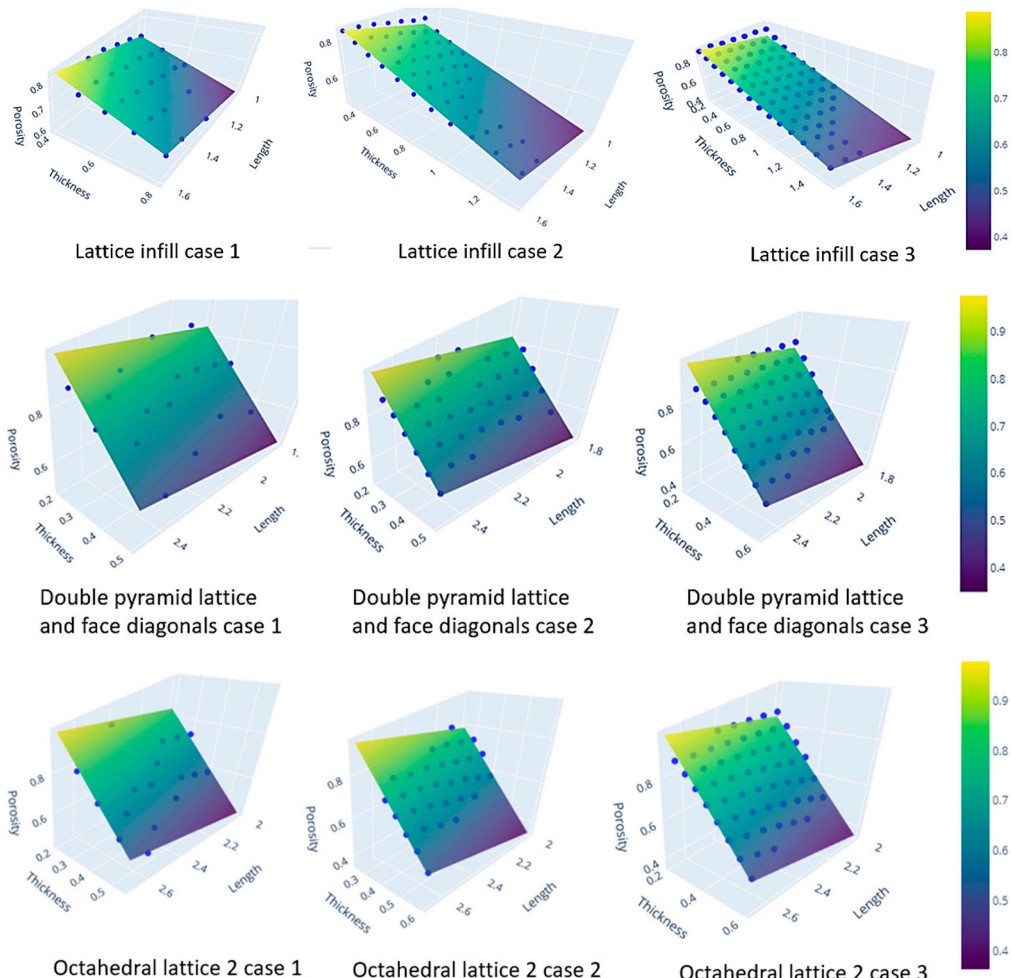

**Figure 5.** Machine learning model representations for all cases.

Figure 6 provides a detailed visualization of the model's accuracy through the inclusion of regression lines. This visual representation shows the difference between the true porosity values and their corresponding predicted values. This offers a comprehensive view of the predictive efficiency of the models at varying sample sizes. It is visible through the figures how closely the model aligns with the actual data across different data points, allowing for a good evaluation of its predictive performance under different scenarios.

The good level of accuracy achieved in predicting the required porosity for the hip implant design significantly streamlines the entire process, classifying it as both efficient and productive. This translates into a great amount of time and effort savings, as it saves the designers from the need for a time-consuming trial-and-error approach to obtain the desired optimal porosity for a specific hip implant design.

It is worth mentioning that the double-pyramid- and octahedral-lattice types showed superior accuracy levels. These levels of accuracy can be attributed to the comparatively more predictable and mathematically tractable inner structures of these lattice types, in contrast to the 3D-lattice infill type. However, the good result lies in the fact that all lattice types achieved an accuracy level of over 96%

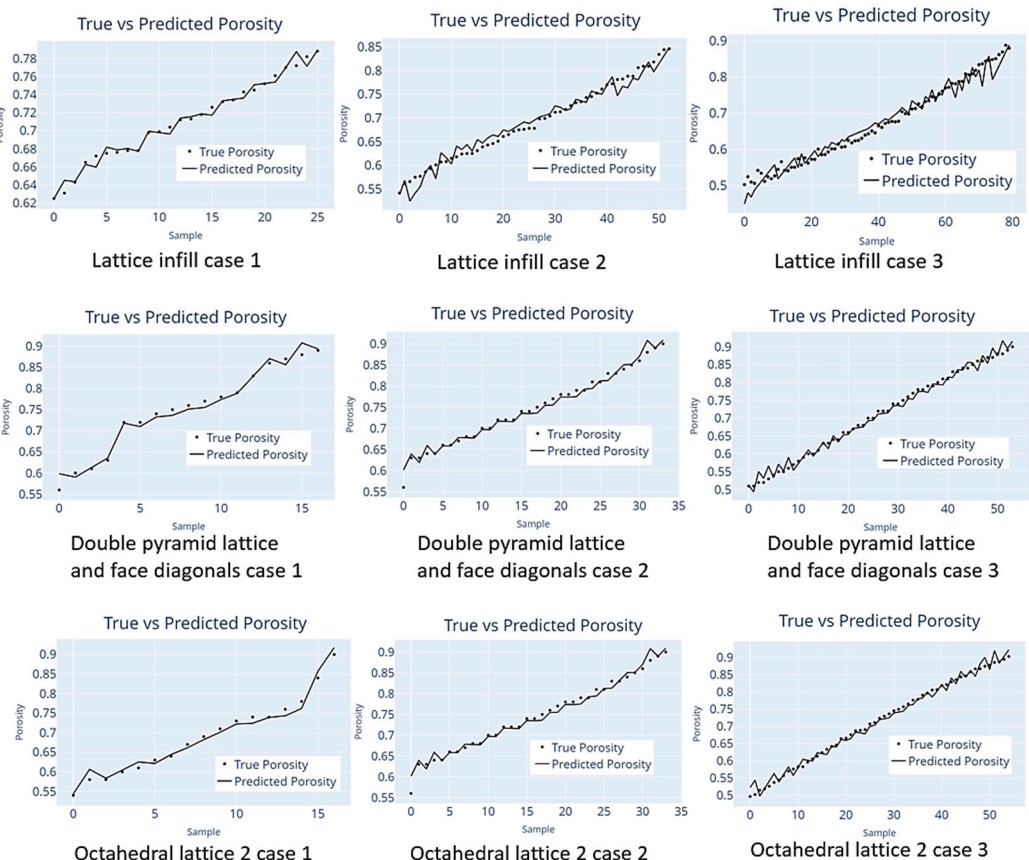

**Figure 6.** Prediction accuracy results for all cases.

## 4. Discussion

Many studies have performed experiments on lattice structures by applying the lattice topology optimization to many designs. In [45], Shengyu et al. showed that for a higher relative density, optimized lattice structures had greater stiffness and strength than the typical lattice design. This research motivated the current study to find a way to create a machine learning algorithm that could predict the porosity of the implant so that the relative density could be easily calculated afterwards.

When starting the computer-aided design (CAD) process for the implant, it is imperative to take into consideration many factors. Firstly, the surface properties play a main role in facilitating osseointegration, ensuring a good integration with the surrounding biological tissue. Additionally, it is essential to engineer the outer layer of the implant body to have a robust mechanical property, guaranteeing durability and stability in practical applications. Simultaneously, the core of the implant must be well-designed to be able to withstand the various loadings that the implant may encounter during its lifespan. Moreover, the chosen geometry should not only meet these functional requirements, but must also be manufacturable with one of the additive manufacturing techniques [46].

Finding the optimum porosity required for the hip implant is crucial to achieve preset mechanical properties. However, the subsequent project might build an inverse model of an algorithm that can predict the perfect variables of unit cells that result in the required porosity. This was performed by predicting the elastic properties in particular, as in [47], and predicting the unit-cell design variables using the mechanical properties as inputs for the algorithm [24].

Using a model to precisely achieve the target mechanical properties for a specific lattice structure design is considered to be an advancement in the process of creating optimized hip implants. In the field of patient-specific hip implant designs, great consideration lies in finding a harmonious match with the biomechanical characteristics of an individual

patient's bone structure. This demands a careful analysis of the bone's unique properties, which helps design an implant that can integrate with the human body as perfectly as possible.

Deformation mechanisms, energy absorption, and the overall mechanical properties of the lattice structures proved to be adjustable by changing their parameters, as shown in [48]. These findings underscore the imperative of finding the means to manipulate the porosity levels within these lattice structures. This ability is crucial as it enables the deliberate fine-tuning of their mechanical attributes, ultimately aiming to attain an optimized configuration that best suits the intended application.

The porosity of the design affects the osseointegration of the implant with the human bone, where a porosity range of 50–90% is believed to be perfect [7]. The porosity also interferes with the tissue regeneration percentage into the implant and affects the elasticity value, where it reduces the Young's modulus into a close-to-human-bone value [41]. Thus, having the ability to predict the porosity has a good effect on the optimization of the hip implant. In brief, the capacity to easily modulate porosity is a tool for achieving the desired mechanical performance of lattice structures.

## 5. Conclusions

In this study, the focus was on refining the porosity of a lattice-structured hip implant design with the leverage of a machine learning algorithm. The primary exploration involved the adjustment of two main parameters: the length and thickness of the lattice structures. The data were collected across all three distinct types of lattice structures. The developed machine learning model demonstrated a predictive capability, achieving an average accuracy level of 98% in estimating the porosity for any lattice type. As noticed, the accuracy surged to 99% for the latter two types, which could be attributed to their shapes conforming to a more predictable pattern compared to the first type.

It is essential to acknowledge that a larger set of training samples leads to more precise predictions. Nevertheless, there exists a balance between the volume of the data and the time and effort invested in its preparation. This equilibrium drove our research to investigate the optimal number of learning points necessary to achieve an acceptable level of accuracy for each model.

The application of machine learning in predicting the design outcomes proved to be highly advantageous, particularly in streamlining the manufacturing process. A future endeavor of this research lies in the development of an inverse model that would enable the estimation of parameters based on the provided target data, representing a challenging yet promising avenue for exploration in the field. Such a trial would improve the way the optimization is approached and further enhance the efficiency of the manufacturing process.

**Author Contributions:** Conceptualization, R.A. and T.M.; Software, R.A.; Formal analysis, R.A.; Supervision, T.M.; Funding acquisition, T.M.; Project administration, T.M.; Validation, R.A.; Methodology, T.M.; software, R.A.; writing, R.A. All authors have read and agreed to the published version of the manuscript.

**Funding:** This research received no external funding.

**Data Availability Statement:** The data presented in this study are available on request.

**Conflicts of Interest:** The authors declare no conflict of interest.

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
