# Peer review of "Optimization of Additively Manufactured and Lattice-Structured Hip Implants Using the Linear Regression Algorithm from the Scikit-Learn Library"

_crystals, doi:10.3390/cryst13101513_

Round 1
Reviewer 1 Report (Previous Reviewer 3)
Overall, the reviewer is satisfied with the authors' revisions.
Author Response
Please see the attachment

Reviewer 2 Report (New Reviewer)
Please read the attachment. Thank you.

Author Response
Please see the attachment.

Reviewer 3 Report (New Reviewer)
Thank you for this contribution. This paper examines the optimization of AM hip implants. The conducted analysis is typically standard and falls within the expected work from such a publication. As such, the authors are invited to properly address the following items:
1. In general, the introduction is light and does not represent state of the art in this domain. The amount of work in this area continues to rise rapidly. The authors are advised to strengthen their literature review section with supplementary material. Perhaps the addition of 1-2 pages can help strengthen this section.
2. The selection of performance metrics in this study is not clear. For example, only R-squared seems to be used. The authors are asked to present their rationale for this metric and not supplement it with other possible metrics. For example, both Alavi et al (2021) [https://doi.org/10.1007/s44150-021-00015-8] and Botchkarev (2019) [https://doi.org/10.28945/4184] present a detailed review of the proper selection of metrics.
3. The distribution of data is not clear nor provided by histograms. It is then hard to visualize how the data is distributed. Please provide such plots.
4. What does this model tell us about the behavior of AM hip implants that we did not know before? In other words, how do the new results match our physics and domain knowledge?
5. How was the training process, a cross validation practice vs. a random split 70/30?
6. Why is the discussion presented after the conclusion section?
.
Author Response
Please see the attachment

This manuscript is a resubmission of an earlier submission. The following is a list of the peer review reports and author responses from that submission.
Round 1
Reviewer 1 Report
The manuscript is not well designed and presented. Novelty is not clear. Machine learning model is used but the architecture is not described in the manuscript. How the training and testing data are generated? lots of details are missing. Therefore, I am against the publication of the manuscript.
Reviewer 2 Report
The work regards an interesting study concerning the influence of the unit-cell geometrical properties on the properties of lattice structures used for hip implant applications. The description of the study is rigorous and clear, and the tone of the manuscript is adapted for a scientific publication. However, before publication, the following aspects should be solved in the revised version of the manuscript:
· Page 3, line 113 – Looking at Figure 1, it is not fully clear if the 2 mm thickness is measured along the x or z direction. Specify this aspect in the manuscript.
· Page 3, line 113 – The list of unit-cell types is reported with a numbering list, whereas in Figure 1, they are labeled with letters (b, c and d). For better clarity, solve this difference.
· Page 3, line 113 – Explain why you are referring to “Octahedral lattice 2” and not simply to “Octahedral lattice”. Is this a lattice structure nomenclature consolidated in the literature?
· Page 4, line 136 – Report one or more literature references for the linear regression machine learning algorithm.
· Page 4, line 149 – Replace “pandas” with “Pandas”.
· Lines 147-167 – Add a Figure with a diagram supporting the algorithm described in this paragraph.
· Page 5, lines 175-179 – The procedure to measure the accuracy is not clear. What is the benchmark value to check the accuracy? Do you compare the predicted value with the value calculated from the CAD model in Figure 1?
· It would be interesting to extend the study with a paragraph showing the influence of the porosity on Young’s modulus since, as explained in paragraph 2, it is an essential parameter for this kind of implant.
· Considering the amplitude of the works discussing this kind of topic in literature, more papers should be added to the references list, such as: https://doi.org/10.1016/j.compstruct.2020.111985; https://doi.org/10.1016/j.euromechsol.2021.104291; https://doi.org/10.1016/j.ijengsci.2019.103198
Author Response
Please see the attachement

Reviewer 3 Report
Briefly speaking, the authors have used public linear regression toolset to build the relationship between beam geometry parameters and its porosity for three lattice structures.
This research suffers from many serious problems, of which some are:
[1] When we build surrogate model, oftentimes we are bascially building an alternative, FAST model to replace the original ones. However, the porosity is supposedly able to be calculated from the beam parameter using some simple math equations. There is thus no necessity for training such a regression model, as the math model is simple and fast (and more importantly, 100% accurate). This shakes the meaningfulness of the entire research.
[2] Quite a few common, public Python library has been used, and a rather simple linear regression model has been implemented in this research. The reviewer did not see any real novelty or authors' contribution in terms of the research methodoly.
[3] The paper also lacks profund and exteneded discussion following the results presentation.
